# The Role of Hypoxia-Sensitive *miRNA181a*, *miRNA199a*, *SIRT1*, and Adiponectin in Diabetes Mellitus Type 2 Development in Obstructive Sleep Apnea Patients [note 1]

**DOI:** 10.3390/ijms26199699

**Published:** 2025-10-05

**Authors:** Filip Franciszek Karuga, Piotr Kaczmarski, Marcin Sochal, Bartosz Szmyd, Greta Veronika Urbonaitė, Szymon Turkiewicz, Piotr Białasiewicz, Agata Gabryelska

**Affiliations:** 1Department of Sleep Medicine and Metabolic Disorders, Medical University of Lodz, 90-419 Lodz, Poland; piotr.kaczmarski@umed.lodz.pl (P.K.); marcin.sochal@umed.lodz.pl (M.S.); szymon.turkiewicz@umed.lodz.pl (S.T.); piotr.bialasiewicz@umed.lodz.pl (P.B.); 2Department of Neurosurgery and Neuro-Oncology, Barlicki University Hospital, Medical University of Lodz, 90-419 Lodz, Poland; bartosz.szymd@umed.lodz.pl; 3Department of Pediatrics, Oncology, and Hematology, Medical University of Lodz, 90-419 Lodz, Poland; 4Clinical Department of Radiotherapy, Radom Oncology Center, 26-600 Radom, Poland; greta.v.urbonaite@gmail.com

**Keywords:** adiponectin, epigenetics, hypoxia, OSA, microRNAs, sirtuin, sleep, metformin

## Abstract

Obstructive sleep apnea (OSA) is a chronic respiratory disorder characterized by intermittent hypoxia and is strongly associated with the development of type 2 diabetes mellitus (T2DM). Despite this link, the molecular mechanisms underlying OSA-related metabolic dysregulation remain incompletely understood. The aim of the study was to investigate the role of hypoxia-sensitive microRNAs, sirtuin 1 (SIRT1), and adiponectin in the metabolic profile of OSA patients, with and without T2DM. A total of 87 participants were stratified into three groups: OSA, OSA + T2DM, and healthy controls. Blood samples were collected in the evening and morning, and after continuous positive airway pressure (CPAP) therapy. Expression levels of miRNAs and *SIRT1* were measured via RT-qPCR; adiponectin was quantified by ELISA. Significantly reduced expression of *miRNA-181a* and *miRNA-199a* was observed in the OSA + T2DM group compared to OSA (*p* = 0.035 and *p* = 0.042, respectively). In contrast, *SIRT1* expression was highest in the OSA + T2DM group (*p* < 0.01), while adiponectin concentrations was lowest in this group and the highest among healthy controls (*p* = 0.001). Despite increased SIRT1 in OSA + T2DM patients, the parallel increase in adiponectin was not observed. Additionally, expression of *SIRT1* was significantly increased in OSA patients who were taking metformin (*n* = 23) vs. patients without metformin (*n* = 32) 77.315 vs. 437.08 (*p* = 0.037). CPAP therapy had significant influence only on *miRNA-181a*—expression was increased after long-term treatment (*p* = 0.047). Increased *miRNA-181a* expression in patients with OSA is related to decreased *SIRT1* expression, which may lead to T2DM development. Surprisingly, the expression of *SIRT1* is significantly higher and expression of hypoxia-sensitive miRNAs is significantly lower in patients with already developed T2DM, which might be explained by metformin intake.

## 1. Introduction

Sleep is a complex physiological process that cannot be reduced to a single dimension. It is shaped by multiple factors, including subjective sleep satisfaction, total sleep duration, sleep latency, frequency of awakenings, and overall sleep quality. Importantly, sleep can also be disrupted by specific sleep disorders, such as insomnia or obstructive sleep apnea (OSA). These disturbances are not only detrimental to sleep architecture itself, but they also trigger a cascade of consequences on various biological levels. Starting from molecular alterations, they extend to cellular, tissue, and hormonal dysregulation, ultimately contributing to a wide range of health outcomes. Among the most significant are cardiovascular diseases, metabolic disorders, and, notably, type 2 diabetes mellitus (T2DM) (see Figure 1).

In this study, we aimed to investigate the role of epigenetic factors, particularly microRNAs in mediating the increased risk of T2DM observed in patients with OSA. By focusing on these molecular mechanisms, our work seeks to bridge the gap between sleep disturbances and their long-term systemic health consequences.

OSA is a chronic disorder marked by repeated episodes of partial or complete upper airway obstruction during sleep, resulting in intermittent hypoxia, hypercapnia, frequent arousals, and fragmented sleep architecture. Epidemiological studies estimate that moderate to severe forms of OSA affect approximately 6% to 17% of adults in the general population. Some reports suggest even higher prevalence rates, reaching up to 23% in women and 49% in men, with a male-to-female ratio ranging from 3:1 to 5:1 [1,2]. The current gold standard for OSA diagnosis is overnight polysomnography (PSG). During PSG, the apnea–hypopnea index (AHI)—representing the number of apneas and hypopneas per hour of sleep—is used to classify severity of OSA: mild (5 ≤ AHI < 15), moderate (15 ≤ AHI < 30), and severe (AHI ≥ 30). OSA-induced sleep disruption contributes to excessive daytime sleepiness, impaired cognitive function, and an increased risk of motor vehicle accidents. The most effective treatment for OSA is continuous positive airway pressure (CPAP) therapy, which maintains airway patency by delivering constant positive air pressure. While CPAP effectively alleviates hypoxia and improves sleep quality, its impact on glycemic control in patients with established T2DM appears limited. Nonetheless, some evidence suggests that CPAP may delay the onset of T2DM in at-risk individuals [3]. Moreover, OSA is recognized as an independent risk factor for both cardiovascular disease and metabolic conditions, including T2DM [4,5]. The study of Mahmood et al. reported a 30.1% prevalence of T2DM among individuals with OSA, compared to 18.6% in those without OSA [6]. Despite the growing advances in research, the precise mechanisms underlying this association remain incompletely understood [7].

This study aims to investigate the potential involvement of microRNAs (miRNAs) in this phenomenon. Initially, hypoxia-sensitive miRNAs were identified. Subsequently, sirtuin 1 (SIRT1), a molecule involved in glucose metabolism and known to be targeted by hypoxia-responsive miRNAs, was selected for further analysis. In addition, adiponectin—a key regulator of glucose metabolism—was not targeted by these miRNAs, was also included.

At the molecular level, hypoxia-inducible factors (HIFs) play a central role in regulating oxygen homeostasis. HIF is a heterodimeric transcription factor consisting of an oxygen-sensitive α-subunit (HIF-1α, HIF-2α, or HIF-3α) and a constitutively expressed β-subunit. HIF-1α, in particular, governs the expression of over 1000 genes involved in processes such as metabolism, angiogenesis, and erythropoiesis [8]. It is consistently found to be upregulated in patients with OSA, suggesting its potential involvement in OSA-related metabolic dysfunction [9].

SIRT1, a nicotinamide adenine dinucleotide (NAD^+^)-dependent deacetylase, is another key regulator implicated in numerous cellular processes including inflammation, apoptosis, and energy metabolism [10]. Recent studies have shown that SIRT1 can modulate hypoxic responses by interacting with HIF-1α, either repressing its transcriptional activity or stabilizing it via direct deacetylation [11,12,13,14]. Notably, *SIRT1* expression is reduced in patients with OSA, while CPAP therapy appears to restore its levels and functional activity [15,16,17]. SIRT1 also plays a crucial role in insulin sensitivity by inhibiting protein tyrosine phosphatase 1B (PTP1B), a negative regulator of insulin signaling, and enhancing glucose-stimulated insulin secretion. Furthermore, studies about T2DM on mouse models revealed an increased expression of PTB1B, which is an inhibitor of the signal transduction and leads to insulin resistance. SIRT1 inhibits PTB1B, leading to sensitization of cells to insulin [18].

Adiponectin is an adipokine with anti-inflammatory and insulin-sensitizing properties. Its concentration is typically reduced in OSA and T2DM patients [19,20]. SIRT1 increases the transcription of the adiponectin gene acting via various factors, including FoxO1, C/EBP alpha, and PPAR gamma; therefore, reduced SIRT1 activity has been shown to downregulate the expression of adiponectin [21,22].

MiRNAs, are nonprotein coding RNAs, small, 18–24 nucleotide molecules that regulate gene expression by binding to target mRNA, which leads to either translational repression or degradation. MiRNAs influence a wide range of biological processes including cell proliferation, apoptosis, stress response, and metabolism [23,24]. Several miRNAs have been shown to exhibit altered expression patterns in OSA patients, particularly in response to intermittent hypoxia [25,26]. For instance, miRNA profiling has revealed differential expression of over 100 miRNAs in the serum of OSA patients compared to controls [27,28]. Some of these findings were later validated in larger cohorts, where changes in specific miRNA levels were observed before and after CPAP therapy [25,29]. Due to financial limitations, the present study focused on a carefully selected subset of miRNAs. Selection criteria included hypoxia responsiveness and predicted targeting of *SIRT1*. The five chosen miRNAs were *miRNA-133, miRNA-181a, miRNA-199a, miRNA-485,* and *miRNA-486*. Among them, *miRNA-181a* and *miRNA-199a* were prioritized based on their robust hypoxia sensitivity, strong target prediction scores for *SIRT1*, and support from previous experimental data [30,31] (see Figure 2).

This study aimed to investigate possible mechanisms involved in distinct changes in oxygen metabolism and their role in T2DM risk among OSA patients through hypoxia-sensitive miRNAs (*miRNA181a, miRNA199a*), SIRT1, and adiponectin. We hypothesized that the intermittent hypoxia characteristic of patients with OSA would alter the expression of *miRNA-181a* and *miRNA-199a*, both of which are hypoxia-sensitive and target SIRT1. Such alterations are expected to lead to a reduction in *SIRT1* expression and subsequent metabolic disturbances. Moreover, the downregulation of SIRT1 is likely to be associated with decreased adiponectin levels, thereby contributing to further metabolic complications.

## 2. Results

### 2.1. Study Groups

The study comprised 87 patients: 26 with OSA, 29 with both OSA and T2DM, and 32 in the control group. There were no significant differences in terms of participants’ sex or total sleep time. We observed significant differences in age and BMI, AHI, desaturation index, fasting glucose, and insulin (see Table 1). In addition, 23 patients from OSA + T2DM group were taking metformin. To evaluate the influence of BMI and age on *miRNA-181a*, *miRNA-199a*, *SIRT1,* and adiponectin, the correlations between the above-mentioned factors were assessed. The only statistically significant correlation was between age and *miRNA199a* (r= −0.22; *p* = 0.047). All other combinations remained unsignificant.

### 2.2. Hypoxia-Sensitive miRNAs: microRNA-181a and miRNA-199a

Significant differences in evening expression levels of *miRNA-181a* and *miRNA-199a*, as well as in the morning concentration of *miRNA-199a*, were observed among the OSA, OSA + T2DM, and control groups, as determined by the Kruskal–Wallis test (see Table 2). Post hoc Dunn’s test revealed that these differences were significant only between the OSA and OSA + T2DM groups, with *p*-values of 0.035, 0.042, and 0.034, respectively. Additionally, in the evening collection, the expression of *miRNA-181a* was lower in group taking metformin compared with patients without metformin 16.41 (IQR: 10.59–21.29) vs. 29.34 (IQR: 16.32–99.17), *p* = 0.003. Similarly, the evening expression of *miRNA 199a* was lower in patients taking metformin 0.45 (IQR: 0.32–1.97) vs. 2.15 (IQR: 0.74–7.17); *p* = 0.006.

### 2.3. Metabolic Guardians: SIRT1 and Adiponectin

Significant differences in evening and morning expression levels of *SIRT1* and serum protein adiponectin levels were observed among the OSA, OSA + T2DM, and control groups, as determined by the Kruskal–Wallis test (see Table 3). Post hoc Dunn’s test revealed that these differences remained significant for adiponectin (*p*-value < 0.001) and *SIRT1* only between the OSA and OSA + T2DM groups, with *p*-value of 0.009 for *SIRT1* evening and 0.006 for *SIRT1* morning expression.

Expression of *SIRT1* was significantly increased in patients suffering from OSA who were taking metformin (*n* = 23) vs. patients without metformin (*n* = 32) 77.315; IQR 30.04–509 vs. 437.08; IQR 14.74–1111.44, respectively (*p* = 0.037). The potential role of hypoxia and microRNAs in pathogenesis T2DM development in OSA patients was also indirectly assessed by correlations. *MiRNA-181a* evening expression presented a weak negative correlation with the *SIRT1* evening and morning expressions (r = −0.28, *p* = 0.010 and r = −0.26, *p* = 0.017, respectively, see Figure 3). Moreover, AHI presented strong positive correlation with both fasting glucose and insulin levels (r = 0.54, *p* < 0.001; r = 0.46, *p* < 0.001, respectively). Regarding the group selection, the only statistically significant correlation was between age and evening expression of *miRNA199a* (r = −0.22; *p* = 0.047; see Figure 4). All the other assessed molecular factors, *miRNA-181a*, *miRNA-199a*, *SIRT1*, and adiponectin, were not significantly correlated with BMI and age.

### 2.4. Results After One-Night CPAP Trial and at Least 3 Month CPAP Treatment

Out of 87 patients only 27 (31.0%) underwent CPAP trial and 21 (24.1%) underwent ≥ 3 months of CPAP treatment. When comparing their microRNA and gene expressions to the initial morning measurement, we observed a significant difference only for miRNA-181 between initial diagnosis and follow-up: 16.168 (IQR: 10.495–32.206) vs. 43.867 (IQR: 22.592–86.208; *p* = 0.048). Furthermore, we observed significant differences in insulin concentration between these time points: 164.15 (IQR: 113.98–303.50) vs. 134.00 (IQR: 98.35–274.00; *p* = 0.030) pmol/L.

## 3. Discussion

In our study, the expression of both *miRNA-199a* and *miRNA-181a* was found to be increased in patients suffering from OSA compared to the OSA + T2DM patients and healthy controls. These observations are not consistent with the Santamaria-Marios et al. study. The validated qPCR expression of morning *miRNA-199a* in the study of Santamaria-Marios et al. was decreased in the OSA patients: a 0.54-fold change (*p* = 0.184). In our study, we observed a 2.4-fold increase in the evening (*p* = 0.047) and a 2.5-fold increase in the morning (*p* = 0.034) compared with healthy controls. Surprisingly the lowest *miRNA-199a* expression was found in OSA + T2DM group—0.25-fold change comparing with OSA group (*p* = 0.047) for evening and 0.21-fold change (*p* = 0.034) for morning measurements. The differences for *miRNA-181a* between the study of Santamaria-Marios et al. and our study were also present. The expression of *miRNA-181a* in the study of Santamaria-Marios et al. was found to be lower in the OSA patients than in controls, a 0.59-fold decrease (*p* = 0.001) compared to a 1.4-fold increase (*p* = 0.041) in our study. Again, *miRNA-181a* expression was the lowest for OSA + T2DM comparing with control group: a 0.46 fold change (*p* = 0.035) [25]. The described differences between the studies may result from the selection of the tested material—in our study we used buffy coat, while in the study of Santamaria-Marios et al., plasma was used. Since both transcription and translation occur intracellularly, the assessment of the intracellular expression levels is an important element to fully understand the described molecular mechanisms. A comparison of the results from both studies may suggest that *miRNA-181a* and *miRNA-199a* are vulnerable to any contamination of samples with blood cells or cell lysis because they present an opposite reaction to hypoxic conditions depending on the material—blood cells or plasma. Any contamination of plasma with blood cells can lead to false results. This may suggest a reduced usefulness of both miRNAs as biomarkers for OSA diagnosis. A more intriguing aspect appears to be the reduced expression of hypoxia-sensitive miRNAs followed by an increased expression of *SIRT1* in patients with OSA and already developed T2DM.

Metformin intake, prolonged exposure to hypoxic environment, or unsuccessful compensation may explain this phenomenon. In the study of Cuyàs et al., it was proven that metformin can be a direct *SIRT 1* activating compound [32]. Moreover, Khowailed et al. in the study conducted on a rats model proved that the administration of metformin increased the *SIRT1* in diabetic animals [33]. In the case of *miRNA-181a*, there are no studies designed in the diabetes context; however, the findings of Oliveras-Ferraros et al. suggesting that metformin downregulates *miRNA181-a* expression is consistent with our study [34]. Therefore, the upregulated *SIRT1* expression mediated via downregulation in *miRNA181-a* can be one of the possible mechanisms of the described effect. In the randomized controlled trial of Zunica et al., it was found that metformin treatment preserved whole-body glucose homeostasis and skeletal muscle mitochondrial function in patients with moderate to severe OSA, indicating a potential role for metformin in preventing the onset of T2DM [35]. On the other hand, despite metformin treatment, the presence OSA symptoms was identified as a significant factor reducing the therapeutic efficacy of metformin in T2DM patients [36].

The negative correlation between the expression of *SIRT1* and *miRNA-181a* described in our study is consistent with the findings of Zhou et al.’s study, which was performed on cell cultures [31]. A decreased expression of *miRNA-181a* leads to *SIRT1* overexpression, which has a beneficial effect on glucose metabolism. In the OSA patients, we observed the opposite [15,37]. Both the increased *miRNA-181a* expression and reduced expression of *SIRT1* in OSA patients may suggest that this is one of the epigenetically mediated causes of the increased incidence of T2DM in the OSA patients; the *SIRT1* expression was the lowest in patients with already developed T2DM.

Due to the inability to assess SIRT1 protein levels directly, we opted to include an additional biomarker in the analysis—one that is detectable in serum (minimizing the risk of contamination), correlated with SIRT1 protein levels, and involved in glucose metabolism. Adiponectin fulfilled all these criteria. Therefore, the measurement of adiponectin may provide an indirect estimation of SIRT1 protein expression. In addition, adiponectin itself plays a key role in glucose homeostasis and the pathogenesis of T2DM, offering the potential for novel insights [38]. Interestingly, despite the highest *SIRT1* and the lowest levels of hypoxia-sensitive miRNAs expression observed in the OSA + T2DM group, adiponectin concentration in this group was the lowest—only slightly higher in the OSA group and markedly elevated in the control group. It may be argued that adiponectin concentration more accurately reflects glucose metabolic status in studied groups than *SIRT1* gene expression. Alternatively, this finding could suggest a diminished effect of chronic hypoxia and metformin treatment on adiponectin levels.

### Study Limitations

Increased BMI and obesity are the main risk factors for both OSA and T2DM development. Due to this, the selection of a control group was challenging. Although efforts were made to recruit age- and BMI-matched groups, the mean BMI differed by as much as 7 kg/m^2^ between the cohorts. This is the main limitation of our study. On the other hand, all patients included in the study were overweight. Furthermore, it was proven that OSA is a risk factor for T2DM development, independent of obesity and other factors [39]. Additionally, in our study, we did not find any statistically significant correlation between analyzed factors and BMI. There was only one statistically significant weak correlation between age and analyzed factors with evening expression of *miRNA-199a.*

Another limitation of this study is the comparison of molecular markers derived from different biological compartments, such as *SIRT1* mRNA expression in buffy coat and circulating adiponectin levels in serum. While buffy coat is a convenient and minimally invasive source of material, its gene expression profile may not directly reflect molecular activity in metabolically active tissues such as the liver, adipose tissue, etc. Therefore, caution is warranted when interpreting associations between buffy coat-derived *SIRT1* expression and systemic markers like adiponectin. This methodological constraint may limit the ability to draw definitive conclusions about tissue-specific regulatory mechanisms, and future studies should consider the inclusion of tissue-specific samples where feasible.

Other limitations of our study include the small number of participants in the prospective part of the study, possible contamination of the buffy coat with plasma components, and generally low levels of *SIRT1* expression.

## 4. Materials and Methods

This study was conducted in the Department of Sleep Medicine and Metabolic Disorders at the Medical University of Lodz from April 2018 until August 2024 as a continuation of a cross-sectional study published in October 2024 with preliminary data [40]. Participants were recruited at the Sleep and Respiratory Disorders Center (Central Clinical Hospital of the Medical University of Lodz). The study was approved by the Bioethical Committee of the Medical University of Lodz (Number of the Bioethical Committee Consent RNN/77/18/KE with an extension KE/1137/20, 13 March 2018). All patients underwent standard physical examination. The diagnosis of T2DM was made based on the patients’ medical documentation or the patients’ clinical factors relying on the recommendations of the Polish Diabetes Society [41]. Patients included in the study needed to meet the following criteria: written informed patient’s consent to participate in the study, age between 30 and 70 years old, and BMI > 20 kg/m^2^ and <45 kg/m^2^. The exclusion criteria were chronic pulmonary disease, history of infection 2 weeks prior and after the PSG examination, active or history of cancer, lifetime history diagnosed sleep disorders other than OSA, clinical depression or other psychiatric disorders, employment in changing shift system, intercontinental flight 2 weeks prior to PSG, abuse of or dependency on alcohol or illegal drugs, caffeine intake > 900 mg per day, use of hypnotic medication or medication known to affect sleep 2 weeks before PSG.

The participants arrived at the sleep laboratory around 9 PM, with a possible variation of 30 min, and underwent a physical examination, which included measuring their body weight, height, heart rate, and blood pressure. During PSG, the following channels were recorded: electroencephalography (C4\A1, C3\A2), electromyography of the chin muscles and anterior tibialis, electrooculography, oronasal airflow (measured with a thermistor), snoring, body position, respiratory movements of the chest and abdomen (measured with piezoelectric sensors), unipolar electrocardiogram, and oxygen saturation (SpO2). The PSG was conducted using an Alice 6 device (Philips-Respironics, Murrysville, PA, USA). Sleep stages were scored based on the 30 s epoch standard. Apnea was defined as a reduction in airflow to less than 10% of baseline for at least 10 s. Hypopnea was characterized by a reduction in airflow of at least 30% for at least 10 s, accompanied by a decrease in SpO2 of over 3% or arousal. The collected data were exported to EDF files and then edited, processed, and analyzed using Neuro-Analyzer v0.23.9. Arousal scoring followed the guidelines of the American Academy of Sleep Medicine. An identical protocol was applied during the first night of CPAP therapy, which functioned as an initial trial preceding long-term treatment. During this session, patients were equipped with a properly fitted mask, and individual therapeutic pressure levels were calibrated. A minimum usage duration of 4 h was required to consider the session effective. For all patients, for whom the CPAP was effective (AHI < 10), an ambulatory CPAP treatment was prescribed. Adherence to the treatment was defined as using the CPAP for at least 5 days a week, and at least 4 h a night. Patients’ adherence was controlled by data stored on the device. Treatment was prescribed independently of the study by an experienced physician.

Blood samples were collected in the evening (at 21:00–21:30, 15 min before lights out), and in the morning (at 6:00–6:30, 15 min after lights on) following the PSG examination, and then centrifuged (see Table 4). However, blood samples after one-night CPAP treatment and during the follow-up were collected only in the morning. The serum and buffy coat were collected and stored at −80 °C. Fasting glucose and insulin levels in the serum from the morning blood were assessed. The insulin resistance was determined using the homeostasis model assessment-estimated insulin resistance (HOMA-IR) formula:$$HOMA-IR=\frac{\left[glucose\;\left(\frac{\mathrm{mg}}{\mathrm{dL}}\right)\right]\times \left[[insulin\;\left(\frac{\mathrm{mU}}{L}\right)\right]}{405}.$$

In the next step, the following molecular and biochemical factors were assessed: *SIRT1* expression at mRNA level, *miRNA-181a* and *miRNA-199a* expression, SIRT1 intracellular protein levels, and serum adiponectin protein level through ELISA. These specific miRNAs were chosen based on the literature and their capability to target *SIRT1* which was additionally validated using microRNA database—miRDB www.mirdb.org (accessed on 3 April 2021) [30].

Total RNA was extracted from the buffy coat using the mirVana PARIS Protein and RNA Isolation System (Invitrogen, Carlsbad, CA, USA) according to the manufacturer’s protocol. The quantity and quality of the RNA were measured with a PicoDrop spectrophotometer (Picodrop Limited, Hinxton, UK). The purified total RNA was immediately used for cDNA synthesis or stored at −80 °C. Reverse transcription was performed using a Maxima First Strand cDNA Synthesis Kit (Thermo Fisher Scientific, Waltham, MA, USA) according to the manufacturer’s recommendations. Expression of sirtuin 1 was assayed using TaqMan^®^ gene expression assays (Thermo Fisher Scientific, Waltham, MA, USA) SIRT1 (Hs1009006_m1), and Actin Beta (ACTB) (Hs1060665_g1) was chosen as endogenous control. All reactions were run on a 7900HT Fast Real-Time PCR System (Applied Biosystems, Foster City, CA, USA) in duplicates differing by <0.5 CT. miRNA reverse transcription was performed using a TaqMan miRNA Reverse Transcription Kit and specific primers (Thermo Fisher Scientific, Waltham, MA, USA) supplied with TaqMan microRNA assays: hsa-miR-181a (assay ID 000480, miRNA sequence: AACAUUCAACGCUGUCGGUGAGU), hsa-miR-199a (assay ID 000498, miRNA sequence: CCCAGUGUUCAGACUACCUGUUC), and RNU6B (assay ID 001093) as an endogenous control. RT reactions were performed according to the manufacturer’s instructions. All reactions were run in duplicates differing by <0.5 CT on 7900HT Fast Real-Time PCR System (Applied Biosystems, Foster City, CA, USA) in 96-well PCR plates. The data were analyzed using SDS 2.3 software and the relative expression of *SIRT1* and miRNA was calculated according to the ΔCt method. To improve readability, *SIRT1* gene expression values were multiplied by 10^−6^ and reported in the normalized format. For quality control purposes, specific genes or microRNA measurements were excluded from the statistical analysis under the following conditions: if the cycle threshold (Ct) values between technical replicates differed by more than 0.5 cycles, or if the Ct value of the endogenous control exceeded 35. These criteria ensured the reliability and consistency of the qPCR results.

For the assessment of the SIRT1 intracellular protein level, the ELISA Human Sirtuin1 (SIRT1), catalog number SEE912Hu (Cloud-Clone Corp, Wuhan, Hubei, China) was used. The total protein level in the buffy coat samples was assessed with a BCA test—an average of 130 mg/mL. Based on the protein levels in the randomly selected samples from each of the tested groups, the ELISA test curve was determined to be 0.78–50 ng/mL. The tests carried out allowed us to draw the conclusion that the concentration of SIRT1 in the tested samples was below the detection level. Cell membrane and nuclear membranes lysis method optimization was applied. The selected methods included sonication on ice 4 times × 5 s with a break of 30 s, RIPA lysis buffer addition to the samples in three different layouts (1:1, 1:4, and 1:9—volume of sample/volume of lysis buffer), and a combination of sonication and buffer lysis. After thawing and membrane lysis, verification of cell membrane integrity by flow cytometry was performed. Due to concentration of SIRT1 protein below the detection level of 0.31 ng/ml the assessment of SIRT1 protein in the buffy coat was withdrawn, as detailed in the preliminary study. Enzyme-linked immunosorbent assay kits were used to assess Human Total Adiponectin/Acrp 30, catalog number DRP300 (Bio-Teche, R&D Systems, Minneapolis, MN, USA) in the serum. The absorbance was measured at a λ = 450 nm wavelength by the absorbance reader (BioTek 800 TS, Agilent Technologies, Santa Clara, CA, USA).

Nominal data were presented as *n* (% of total). Dependencies between nominal variables were tested using the Chi-square test, Chi-square test with Yates’s correction, or Fisher’s exact test, depending on the sample size of the smallest subgroup (cut-off points were 15 and 5, respectively). The normality of continuous variables was assessed using the Shapiro–Wilk test. Normally distributed data were presented as means with standard deviations (SD) and analyzed using Student’s t-test for dependent or independent variables, or ANOVA for comparisons involving more than two groups. In cases of non-normal distribution, data were presented as medians with interquartile ranges (IQR: 1st quartile—3rd quartile) and analyzed using the Mann–Whitney U test, Wilcoxon test, or Kruskal–Wallis test for comparisons involving more than two groups. Appropriate post hoc tests were performed for ANOVA and Kruskal–Wallis tests. Correlations between continuous variables were evaluated using Spearman’s rank correlation coefficient. A *p*-value of <0.05 was considered statistically significant unless otherwise stated. Statistical analyses were performed using Statistica 13.1 PL (TIBCO Polska, Krakow, Poland).

## 5. Conclusions

To conclude, increased *miRNA-181a* expression in patients with OSA is related to decreased *SIRT1* expression, which may lead to increased insulin resistance and T2DM development. Surprisingly, the expression of *SIRT1* is significantly higher and expression on hypoxia-sensitive miRNAs is significantly lower in patients with already developed T2DM. However, increased *SIRT1* expression in OSA + T2DM patients is not accompanied by increased levels of beneficial adiponectin. Metformin intake in OSA + T2DM is the most possible cause of this phenomenon. In the group treated with CPAP, a significant increase in *miRNA-181a* expression and reduced levels of insulin was observed. *SIRT1* and adiponectin levels did not present statistically significant change. Addition of metformin to the treatment of severe OSA in patients with high risk of TDM2 development could possibly reduce that risk.

## Figures and Tables

**Figure 1 ijms-26-09699-f001:**
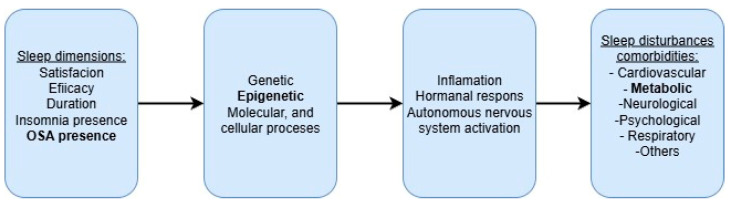
Conceptual framework linking sleep dimensions, sleep disorders, and sleep-related comorbidities.

**Figure 2 ijms-26-09699-f002:**
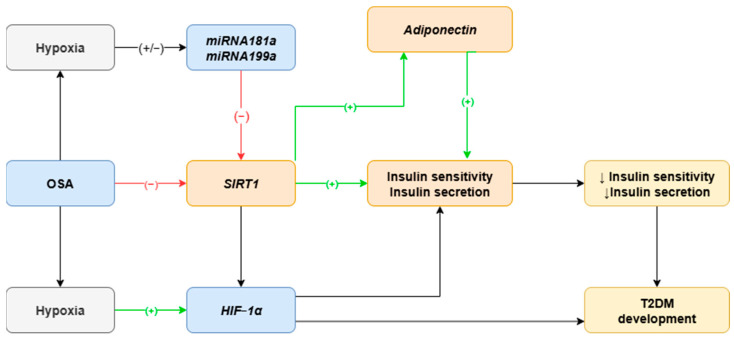
Role of HIF-1α, SIRT1, adiponectin, *miRNA199a*, and *miRNA181a* in T2DM development in OSA patients. SIRT1 increases insulin sensitivity and insulin secretion. Level of SIRT1 is lower in patients suffering from OSA, which is responsible for insulin resistance and insulin-reduced secretion followed by T2DM development. Moreover, SIRT1 increased expression is followed by an increased adiponectin concentration. Hypoxia-sensitive miRNAs may have influence on SIRT1. In addition, SIRT1 may act via HIF-1α, which is responsible for activation of vast number of genes. OSA—obstructive sleep apnea; T2DM—type 2 diabetes mellitus; SIRT1—sirtuin 1; miRNA—microRNA; HIF-1α—hypoxia-inducible factor-1α; ↓, (-), red color arrow—downregulation; (+), green color arrow—upregulation.

**Figure 3 ijms-26-09699-f003:**
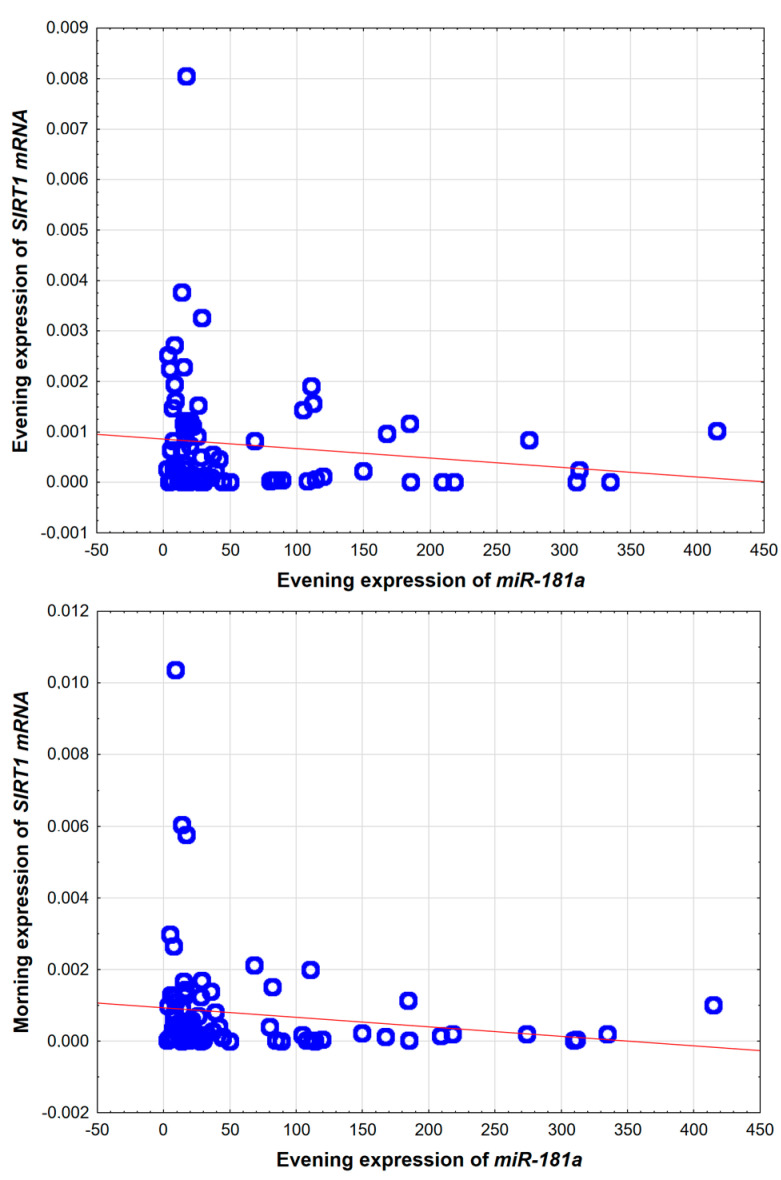
Correlations between evening expression of *miRNA-181a* and evening and morning expression of *SIRT1*. *SIRT1*—Sirtuin 1. Red line—trendline.

**Figure 4 ijms-26-09699-f004:**
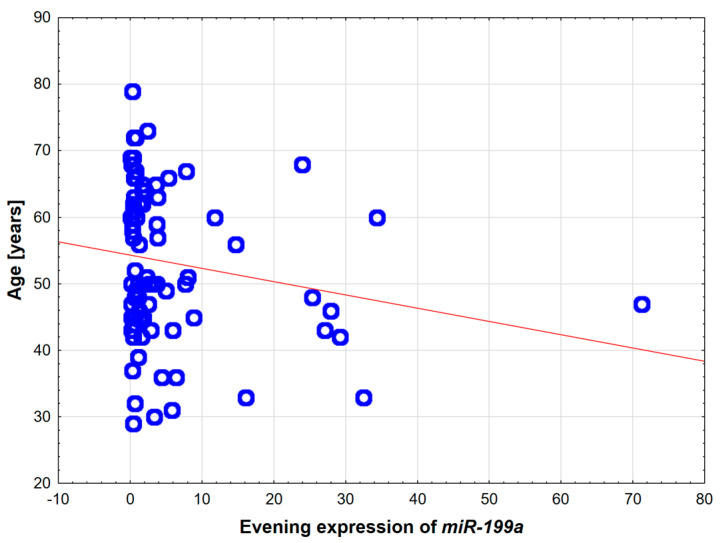
Correlation between age and evening expression of *miRNA199a*—the only significant correlation (*p* = 0.47) between analyzed molecular factors and group characteristics. Red line—trendline.

**Table 1 ijms-26-09699-t001:** Demographic and clinical characteristics of study and control group. OSA—obstructive sleep apnea; T2DM—diabetes mellitus type 2; AHI—apnea–hypopnea index; BMI—body mass index.

	OSA	OSA + T2DM	Control Group	*p*-Value
N	26 (29.9%)	29 (33.3%)	32 (36.8%)	N.A.
Female sex	4 (15.4%)	5 (17.2%)	7 (21.9%)	0.441
Age [years old]	50.5 (IQR: 45.25–62)	58 (IQR: 49–67)	47 (IQR: 42–54)	0.004
BMI [kg/m^2^]	31.794 (IQR: 28.091–36.499)	35.891 (IQR: 32.084–39.246)	28.532 (IQR: 26.967–31.868)	<0.001
AHI [incidents/h]	40.35 (IQR: 36.25–59.375)	55.1 (IQR: 28.4–80.3)	2.1 (IQR: 1.175–4.2)	<0.001
Desaturation index	43 (IQR: 24–56.3)	62 (IQR: 33.3–80)	2 (IQR: 1–3.55)	<0.001
Total sleep time [h]	6.3 (IQR: 5.2–7)	5.88 (IQR: 5.201–7)	6.2 (IQR: 5.4–6.5)	0.771
Glucose [mmol/L]	5.755 IQR: 5.32–5.957	6.89 IQR: 6.35–7.46	5.31 IQR: 5.177–5.565	<0.001
Insulin [pmol/L]	137 IQR: 102.375–168.175	151.7 IQR: 110.2–212.6	68.05 IQR: 59.7–92.575	<0.001

**Table 2 ijms-26-09699-t002:** Comparison of hypoxia-sensitive miRNAs expression in the evening and in the morning between the groups. OSA—obstructive sleep apnea; T2DM—diabetes mellitus type.

	OSA *n* = 26	OSA + T2DM *n* = 29	Control *n* = 32	*p*-Value
*miRNA-181a * evening	36.746 IQR: 18.11–108.35	16.78 IQR: 13.46–27.546	25.619 IQR: 7.002–114.02	0.041
*miRNA-199a * evening	2.988 IQR: 1.01–8.23	0.752 IQR: 0.398–2.264	1.237 IQR: 0.481–5.27	0.047
*miRNA-181a * morning	34.301 IQR: 10.96–81.19	18.279 IQR: 11.335–24.91	15.643 IQR: 9.31–144.19	0.345
*miRNA-199a * morning	2.788 IQR: 0.741–7.886	0.591 IQR: 0.301–1.587	1.137 IQR: 0.468–7.342	0.034

**Table 3 ijms-26-09699-t003:** Comparison of *SIRT1* and adiponectin expression in the evening and in the morning between the groups. OSA—obstructive sleep apnea; T2DM—diabetes mellitus type; *—multiplication.

	OSA *n* = 26	OSA + T2DM *n* = 29	Control *n* = 32	*p*-Value
*SIRT1* [* 10^−6^] evening	44.12 IQR: 27.88–267.96	506.60 IQR: 165.5–1176.7	323.08 IQR: 32.50–1136.35	0.012
*SIRT1* [* 10^−6^] morning	84.44 IQR: 48.49–156.49	553.75 IQR: 206.18–1254.58	214.86 IQR: 46.02–1055.07	0.008
Adiponectin evening [ng/mL]	3849.005 IQR: 2812.635–6048	3170.35 IQR: 2658.19–4366.99	5907.54 IQR: 4142.64–8347.63	0.001
Adiponectin morning [ng/mL]	3327.035 IQR: 2678.85–5013.067	2953.49 IQR: 2296.637–3915.38	5207.775 IQR: 3604.97–7138.832	0.001

**Table 4 ijms-26-09699-t004:** Sample type used for the analysis of hypoxia-sensitive miRNAs, SIRT1, and adiponectin, along with a description of their expression profiles.

Biomolecule	Sample Type	Expression
*miRNA-181a*	buffy coat	yes
*miRNA199a*	buffy coat	yes
*SIRT1* mRNA	buffy coat	yes, but low
SIRT1 protein	buffy coat	no
Adiponectin	serum	yes

## Data Availability

The original data presented in the study are openly available in RepOD—at https://doi.org/10.18150/FPDTPE.

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
