# Peer review of "The Role of Hypoxia-Sensitive *miRNA181a*, *miRNA199a*, *SIRT1*, and Adiponectin in Diabetes Mellitus Type 2 Development in Obstructive Sleep Apnea Patients [Author-notes fn1-ijms-26-09699]"

_ijms, 2025, doi:10.3390/ijms26199699_

Round 1

Reviewer 1 Report

Comments and Suggestions for Authors

This study investigates the changes in hypoxia-sensitive miRNAs, SIRT1, and adiponectin in OSA patients with and without T2DM, which is an interesting exploration. I believe the data itself is valuable. However, the rationale behind the study design is not sufficiently elaborated, which presents a challenge for interpreting the results and understanding their physiological significance.

Firstly, what is the specific biological justification for selecting these three molecules? Are they integral components of a shared signaling pathway, positioned at key nodes? Proposing a pre-defined hypothesis that connects all three would significantly strengthen the study's logical framework.

Secondly, methodological clarity is paramount. From which compartment were the expression levels of miRNAs and SIRT1 measured—serum, plasma, or peripheral blood mononuclear cells (PBMCs)? Adiponectin was quantified in serum via ELISA. The biological significance of measuring molecules from different compartments and their direct comparability requires careful discussion. For instance, can SIRT1 mRNA levels in PBMCs directly reflect its activity in metabolically relevant tissues like the liver or adipose tissue, and thereby be linked to serum adiponectin concentrations? The authors should clarify the sample sources and discuss the potential limitations of this methodological choice for the interpretation of their results.

Author Response

This study investigates the changes in hypoxia-sensitive miRNAs, SIRT1, and adiponectin in OSA patients with and without T2DM, which is an interesting exploration. I believe the data itself is valuable. However, the rationale behind the study design is not sufficiently elaborated, which presents a challenge for interpreting the results and understanding their physiological significance.

Thank you for your insightful comments. We made substantial changes in the manuscript following your instructions.

Firstly, what is the specific biological justification for selecting these three molecules? Are they integral components of a shared signaling pathway, positioned at key nodes? Proposing a pre-defined hypothesis that connects all three would significantly strengthen the study's logical framework.

The first paragraph of the introduction was reorganized. A new paragraph with a framework of the paper was added: “This study aims to investigate the potential involvement of microRNAs (miRNAs) in this phenomenon. Initially, hypoxia-sensitive miRNAs were identified. Subsequently, sirtuin 1 (SIRT1), a molecule involved in glucose metabolism and known to be targeted by hypoxia-responsive miRNAs, was selected for further analysis. In addition, adiponectin—a key regulator of glucose metabolism not targeted by these miRNAs.”

Secondly, methodological clarity is paramount. From which compartment were the expression levels of miRNAs and SIRT1 measured—serum, plasma, or peripheral blood mononuclear cells (PBMCs)? Adiponectin was quantified in serum via ELISA.

To improve methodological clarity, the additional table – Table 3, was added.

The biological significance of measuring molecules from different compartments and their direct comparability requires careful discussion. For instance, can SIRT1 mRNA levels in PBMCs directly reflect its activity in metabolically relevant tissues like the liver or adipose tissue, and thereby be linked to serum adiponectin concentrations? The authors should clarify the sample sources and discuss the potential limitations of this methodological choice for the interpretation of their results.

The new paragraph in Discussion – limitations has been added: “Another limitation of this study is the comparison of molecular markers derived from different biological compartments, such as SIRT1 mRNA expression in buffy coat and circulating adiponectin levels in serum. While buffy coat are a convenient and minimally invasive source of material, their gene expression profile may not directly reflect molecular activity in metabolically active tissues such as the liver, adipose tissue etc. Therefore, caution is warranted when interpreting associations between buffy coat-derived SIRT1 expression and systemic markers like adiponectin. This methodological constraint may limit the ability to draw definitive conclusions about tissue-specific regulatory mechanisms, and future studies should consider the inclusion of tissue-specific samples where feasible.”

Reviewer 2 Report

Comments and Suggestions for Authors

Major Comments

While the results were generally well presented, the authors appear to have overlooked the effects of metformin. Specifically, the abstract does not mention any findings regarding the effects of metformin, and this needs to be discussed in more detail in the Discussion section. The authors recommend implementing the suggested modifications.

Minor Comments

  1. When using abbreviations in the main text, they should be spelled out at their first mention. Spell out CPAP in the abstract
  2. Need to add appropriate references frequently to each sentence, for example “Some reports suggest even higher prevalence rates, 39 reaching up to 23% in women and 49% in men, with a male-to-female ratio ranging from 40 3:1 to 5:1.”
  1. Remove “protein tyrosine phosphatase 1B” on line 73. You already defined abbreviate of PTP1B on line 71
  2. Need to be consistent between HIF-1α and HIF-1alfa
  3. Need to adjust the size of table 2. The table is displayed broken in the text.
  4. Need to adjust the title of Y axis in Figure 2 for adiponectin
  5. Need to provide the correlation data in Figures like figure 2 for “between age and evening miRNA199a”. Just mentioning it in the text is not enough.
  6. Spell out PSG, miRDB
  7. Need to correct 0,31 ng/ml to 0.31 ng/ml on line 351
  8. Need to correct miRNA-181a to miRNA-181α. In addition, the author frequently used "a" instead of "α" in several places throughout the text. This needs to be corrected.

Comments on the Quality of English Language

Need to be improved 

Author Response

Thank you for your insightful and detailed comments. We made substantial changes in the manuscript following your instructions. The manuscript has undergone a thorough language revision. The English was carefully checked and corrected by a professional language expert specialized in academic writing. We believe the current version meets the standards of scientific English.

Major Comments

While the results were generally well presented, the authors appear to have overlooked the effects of metformin. Specifically, the abstract does not mention any findings regarding the effects of metformin, and this needs to be discussed in more detail in the Discussion section. The authors recommend implementing the suggested modifications.

The abstract and discussion section was reorganized. Additional information about metformin was added. The new keyword “metformin: was added. Moreover, the paragraph in discussion section regarding metformin was separated and extended: “Metformin intake, prolonged exposure to hypoxic environment or unsuccessful com-pensation may explain this phenomenon. In the study of Cuyàs et al. it was proved that metformin can be a direct SIRT 1 activating compound [30]. Moreover, Khowailed et al. in the study conducted on a rats model proved that the administration of met-formin increased the SIRT1 in diabetic animals [31]. In case of miRNA-181a there are no studies designed in the diabetes context, however, the findings of Oliveras-Ferraros et. al suggested that metformin downregulates miRNA181-a expression are consistent with our study [32]. Therefore, the upregulated SIRT1 expression mediated via down-regulation in miRNA181-a can be one of possible mechanisms of the described effect. In the randomized controlled trial of Zunica et al., it was found that metformin treatment preserved whole-body glucose homeostasis and skeletal muscle mitochondrial function in patients with moderate to severe OSA, indicating a potential role for metformin in preventing the onset of type 2 diabetes in this population [33]. On the other hand, despite metformin treatment, the presence OSA symptoms was identified as a significant factor reducing the therapeutic efficacy of metformin in T2DM patients [34]. 

 Minor Comments

When using abbreviations in the main text, they should be spelled out at their first mention. Spell out CPAP in the abstract

Done

Need to add appropriate references frequently to each sentence, for example “Some reports suggest even higher prevalence rates, 39 reaching up to 23% in women and 49% in men, with a male-to-female ratio ranging from 40 3:1 to 5:1.”

Done

Remove “protein tyrosine phosphatase 1B” on line 73. You already defined abbreviate of PTP1B on line 71

Done

Need to be consistent between HIF-1α and HIF-1alfa

Done

Need to adjust the size of table 2. The table is displayed broken in the text.

Done

Need to adjust the title of Y axis in Figure 2 for adiponectin

Thank you for noticing, the wrong figure was uploaded.  The Figure 2 was corrected, the B part was changed to match the manuscript text and fulfil the visual standards.

Need to provide the correlation data in Figures like figure 2 for “between age and evening miRNA199a”. Just mentioning it in the text is not enough.

The new figure – Figure 3 was added.

Spell out PSG, miRDB

Done

Need to correct 0,31 ng/ml to 0.31 ng/ml on line 351

Done

Need to correct miRNA-181a to miRNA-181α. In addition, the author frequently used "a" instead of "α" in several places throughout the text. This needs to be corrected.

We assessed miRNA-181a, not α. We corrected all “a” in places regarding HIF1.

Round 2

Reviewer 1 Report

Comments and Suggestions for Authors

I have reviewed the revised manuscript and the authors' responses. Although some text has been added, I am disappointed to find that my main concerns about the rationale and methodological coherence of this study have not been properly addressed. The revisions lack the deeper reflection needed to strengthen the logical foundation of the paper.

My major criticism was that the choice of these molecules—miRNAs, SIRT1, and adiponectin—seemed arbitrary, without a clear pre-specified biological hypothesis connecting them. The authors’ reply still does not solve this problem.

Even if I were to provide additional feedback, I would likely end up raising the same concerns I highlighted in my initial review. I expect the authors to address these points seriously.

Author Response

Comment: 

I have reviewed the revised manuscript and the authors' responses. Although some text has been added, I am disappointed to find that my main concerns about the rationale and methodological coherence of this study have not been properly addressed. The revisions lack the deeper reflection needed to strengthen the logical foundation of the paper.

My major criticism was that the choice of these molecules—miRNAs, SIRT1, and adiponectin—seemed arbitrary, without a clear pre-specified biological hypothesis connecting them. The authors’ reply still does not solve this problem.

Even if I were to provide additional feedback, I would likely end up raising the same concerns I highlighted in my initial review. I expect the authors to address these points seriously.

Response: 

We appreciate the reviewer’s constructive comments and understand the concern regarding the need for a clearer and more coherent rationale linking the studied molecules and the lack of a pre-specified biological hypothesis. We acknowledge that in the initial and revised versions of the manuscript, this connection was not articulated with sufficient depth.

Our biological hypothesis was not arbitrary, but rather based on converging evidence from several studies:

  1. miRNA-181a and miRNA-199a have been repeatedly reported as hypoxia-sensitive regulators, making them highly relevant to the intermittent hypoxia observed in OSA:

“Several miRNAs have been shown to exhibit altered expression patterns in OSA pa-tients, particularly in response to intermittent hypoxia [25,26]. For instance, miRNA profiling has revealed differential expression of over 100 miRNAs in the serum of OSA patients compared to controls [27,28]. Some of these findings were later validated in larger cohorts, where changes in specific miRNA levels were observed before and after CPAP therapy [25,29]. Due to financial limitations, the present study focused on a carefully selected subset of miRNAs. Selection criteria included hypoxia responsive-ness and predicted targeting of SIRT1. The five chosen miRNAs were miRNA-133, miRNA-181a, miRNA-199a, miRNA-485, and miRNA-486”

  1. Both miRNAs directly target SIRT1, a key metabolic regulator that modulates mitochondrial function, oxidative stress response, and insulin sensitivity:

“Among them, miRNA-181a and miRNA-199a were prioritized based on their robust hypoxia sensitivity, strong target prediction scores for SIRT1, and support from previous experimental data”

“Recent studies have shown that SIRT1 can modulate hypoxic responses by interacting with HIF-1α, either repressing its transcriptional activity or stabilizing it via direct deacetylation [11–14]. Notably, SIRT1 expression is reduced in patients with OSA, while CPAP therapy appears to restore its levels and functional activity [15–17]. SIRT1 also plays a crucial role in insulin sensitivity by inhibiting protein tyrosine phosphatase 1B (PTP1B), a negative regulator of insulin signaling, and enhancing glucose-stimulated insulin secretion.”

“The negative correlation between the expression of SIRT1 and miRNA-181a described in our study is consistent with the findings of Zhou et al.’s study, which was performed on cell cultures [35].” – another study describing the relationship between SIRT1 and miRNA-181a.

  1. Reduced SIRT1 activity has been shown to decrease adiponectin expression, an adipokine strongly implicated in glucose and lipid homeostasis, and in the pathogenesis of type 2 diabetes:

“Adiponectin is an adipokine with anti-inflammatory and insulin-sensitizing proper-ties.  Its concentration is typically reduced in OSA and T2DM patients[19,20]. SIRT1 increases the transcription of the adiponectin gene acting via various factors, includ-ing: FoxO1, C/EBP alpha and PPAR gamma, therefore reduced SIRT1 activity has been shown to downregulate the expression of adiponectin [21,22].”

Moreover, all of these relationships are presented in a simplified form in Figure 2.

In order to address the concern regarding the lack of a clear conceptual rationale, we have substantially revised the Introduction.

  1. We added a new section presenting a broader biological framework linking sleep disturbances to their comorbidities, highlighting the specific components on which we focused. Additionally, we included a new figure (Figure 1) to illustrate this framework.

“ Sleep is a complex physiological process that cannot be reduced to a single dimension. It is shaped by multiple factors, including subjective sleep satisfaction, total sleep du-ration, sleep latency, frequency of awakenings, and overall sleep quality. Importantly, sleep can also be disrupted by specific sleep disorders, such as insomnia or obstructive sleep apnea (OSA). These disturbances are not only detrimental to sleep architecture itself, but they also trigger a cascade of consequences on various biological levels. Starting from molecular alterations, they extend to cellular, tissue, and hormonal dysregulation, ultimately contributing to a wide range of health outcomes. Among the most significant are cardiovascular diseases, metabolic disorders, and, notably, type 2 diabetes.

In this study, we aimed to investigate the role of epigenetic factors, particularly microRNAs in mediating the increased risk of type 2 diabetes observed in patients with obstructive sleep apnea. By focusing on these molecular mechanisms, our work seeks to bridge the gap between sleep disturbances and their long-term systemic health consequences.”

  1. We extended the section about adiponectin: “ Adiponectin is an adipokine with anti-inflammatory and insulin-sensitizing properties. Its concentration is typically reduced in OSA and T2DM patients[19,20]. SIRT1 increases the transcription of the adiponectin gene acting via various factors, including: FoxO1, C/EBP alpha and PPAR gamma, therefore reduced SIRT1 activity has been shown to downregulate the expression of adiponectin [21,22]”
  2. A new citation has been added: Rane, S.; He, M.; Sayed, D.; Vashistha, H.; Malhotra, A.; Sadoshima, J.; Vatner, D.E.; Vatner, S.F.; Abdellatif, M. Downregulation of miR-199a Derepresses Hypoxia-Inducible Factor-1α and Sirtuin 1 and Recapitulates Hypoxia Preconditioning in Cardiac Myocytes. Additionally, the paper by Zhou et al. (citations 30 and 31) has been referenced in the text: “Among them, miR-181a and miR-199a were prioritized based on their robust hypoxia sensitivity, strong target prediction scores for SIRT1, and support from previous experimental data [30,31]”, clarifying which experimental data we are talking about.

  1. The last, but most important, change is an addition of a clear pre-specified hypothesis: “We hypothesized that the intermittent hypoxia characteristic of patients with obstructive sleep apnea (OSA) would alter the expression of miRNA-181a and miRNA-199a, both of which are hypoxia-sensitive and target SIRT1. Such alterations are expected to lead to a reduction in SIRT1 expression and subsequent metabolic disturbances. Moreover, the downregulation of SIRT1 is likely to be associated with decreased adiponectin levels, thereby contributing to further metabolic complications.”

We thank the reviewer for highlighting this gap and believe that these revisions will significantly improve the clarity and scientific foundation of our work. If necessary, we are willing to make any additional changes suggested by the reviewer.

Reviewer 2 Report

Comments and Suggestions for Authors All issues raised have been resolved. There are no further questions.

Author Response

Thank you

Round 3

Reviewer 1 Report

Comments and Suggestions for Authors

The authors have done an excellent job in revising the manuscript.

All my previous concerns have been fully addressed.

The paper is now much stronger and I recommend it for acceptance.